# An Evaluation of Food and Nutrient Intake among Pregnant Women in The Netherlands: A Systematic Review

**DOI:** 10.3390/nu15133071

**Published:** 2023-07-07

**Authors:** Sovianne ter Borg, Nynke Koopman, Janneke Verkaik-Kloosterman

**Affiliations:** National Institute for Public Health and the Environment, 3721 BA Bilthoven, The Netherlands; sovianne.ter.borg@rivm.nl (S.t.B.); nynke.koopman@rivm.nl (N.K.)

**Keywords:** pregnancy, dietary assessment, nutritional status, early life, recommendations

## Abstract

Nutritional deficiencies during pregnancy can have serious consequences for the health of the (unborn) child. This systematic review provides an updated overview of the available food and nutrient intake data for pregnant women in The Netherlands and an evaluation based on the current recommendations. Embase, MEDLINE, and national institute databases were used. Articles were selected if they had been published since 2008 and contained data on food consumption, nutrient intake, or the status of healthy pregnant women. A qualitative comparison was made with the 2021 Dutch Health Council recommendations and reference values. A total of 218 reports were included, representing 54 individual studies. Dietary assessments were primarily performed via food frequency questionnaires. Protein, vitamin A, thiamin, riboflavin, vitamin B_6_, folate, vitamin B_12_, vitamin C, iron, calcium, and magnesium intakes seemed to be adequate. For folate and vitamin D, supplements were needed to reach the recommended intake. The reasons for concern are the low intakes of fruits, vegetables, and (fatty) fish, and the intakes of alcohol, sugary drinks, and salt. For several foods and nutrients, no or limited intake data were found. High-quality, representative, and recent data are needed to evaluate the nutrient intake of pregnant women in order to make accurate assessments and evaluations, supporting scientific-based advice and national nutritional policies.

## 1. Introduction

Nutritional deficiencies during pregnancy can have serious consequences for the health of the (unborn) child [1,2,3,4]. Iron deficiency, for instance, leads to anemia, which in turn is associated with impaired fetal development, preterm delivery, and a low birth weight [3]. Another well-known example is folate deficiency, which is associated with anemia, but also with neural tube defects, which can lead to infant mortality and serious disabilities [3]. Adequate nutrition is therefore vital during pregnancy as well as preconceptionally. Previous research in the USA has indicated that significant numbers of pregnant women do not meet the recommendations for multiple vitamins and minerals [5].

Pregnant women may follow the dietary guidelines for the general female population [6]. However, for some foods and nutrients, pregnant women have specific needs that are related to body maintenance, tissue growth, the development of the fetus, or food safety. Examples include a higher requirement for folic acid and iodine, and on the other hand, the prevention of excessive vitamin A intake and the advice not to consume alcohol [3]. 

Because of these differences, there are specific dietary recommendations and dietary reference values for pregnant women. For example, the Nordic Nutrition Recommendations 2012 and the Dietary Guidelines for Americans 2020–2025 include specific recommendations for pregnant women [7,8]. 

In The Netherlands, the Dutch Dietary Guidelines 2015 describe a healthy diet for the general population. These guidelines were not specific for pregnant women. Several organizations have, however, published recommendations for pregnant women, such as the use of folic acid supplements. To harmonize these recommendations, and to support the scientific basis, the Health Council of The Netherlands evaluated the recommendations in light of new scientific developments. In 2021, the Health Council published dietary recommendations (see Table 1) and complementary dietary reference values specifically for pregnant women (see Table 2) [6,9]. 

It is important to understand whether women comply with these dietary recommendations and reference values. These insights serve as a basis for developing effective intervention strategies and policies to prevent potential health risks. In 2021, 179,441 children were born in The Netherlands, a decline compared to previous years [13]. The average number of children per women was 1.62 [13]. For this future generation, a good start in life is essential. To gain an insight into food and nutrient intake during pregnancy in The Netherlands, we previously performed a systematic review of the available nutritional data during the first 1000 days of life (i.e., from conception up to 2 years of age) [14]. At that time, the dietary recommendations and reference values of the Dutch Health Council were not yet available. The current review will provide an updated overview of the available data for pregnant women and will make a comparison with the new recommendations of the Dutch Health Council. 

The aim of the current review is to determine whether the nutritional intake of pregnant women in The Netherlands is in line with the Dutch dietary recommendations and reference values.

## 2. Materials and Methods

This systematic review is reported according to the Preferred Reporting Items for Systematic Reviews and Meta-Analyses (PRISMA) 2020 guideline [15].

### 2.1. Literature Search

In 2019, a systematic review was published on the food consumption, nutrient intake, and nutrient status during the first 1000 days of life [14]. Since then, this living systematic review has been updated via monthly searches. The electronic database Embase was searched, which includes records from the MEDLINE (Pubmed) and Embase databases. A search string was created based on the PICO model. The population (P) of interest was the first 1000 days of life, including pregnant women, mothers during the breastfeeding period, and children up to two years of age. Included were healthy populations living in The Netherlands. Intervention (I) studies were excluded, except when baseline data or data from a healthy control group were available. No comparison (C) was included due to the nature of this systematic review. The outcomes of interest (O) were data on food and nutrient intake, dietary supplement use, and biochemical nutrient status markers. Subsequently articles were selected during the screening phase which contained data from pregnant women. For the full details on the search string, see Appendix A, Table A1. Emtree index terms were used and exploded. The search filters for humans, publication date, and publication type were used. Articles published between January 2008 and January 2023 were included. The starting date of 2008 was chosen as previous recommendations from the Dutch Healthy Council included literature that was published before and up to 2008 [16,17,18,19]. Scientific congress posters and abstracts were excluded. No language restrictions were used. In addition to the electronic database, relevant reports from Dutch institutes and non-peer-reviewed Dutch articles were retrieved. Parallel to the literature search, a second search was conducted on vitamin D and folate intake and status among women of childbearing age (see Appendix A, Table A2). Articles relevant for the current study were identified (*n* = 57) and added to the review. The database and publication date ranges of this parallel search were similar to the main literature search.

### 2.2. Screening and Extraction

Article screening was performed in duplicate by two independent researchers (N.K. and S.t.B.) based on predefined exclusion criteria. A third reviewer (J.V.-K.) was consulted in case of uncertainty about the inclusion of an article. The exclusion criteria were as follows: published before 2008; not containing Dutch data; population with medical illness or premature infants; population that was not pregnant, breastfeeding, or included children with a mean age above 2 years; no data on food consumption, nutrient intake, nutrient status, or supplement use; intervention studies without a healthy control group or baseline data; paternal preconception data; case studies; duplicate data; and articles for which the full text could not be retrieved. Exclusion criteria for the parallel search on women of childbearing age were similar to the main search, with two exceptions. Articles needed to include information on vitamin D of folate/folic acid, and women with a mean age younger than 20 years or older than 45 years were excluded. 

The articles were subsequently divided amongst the researchers (N.K. and S.t.B.) to extract the study characteristics. The following study characteristics were extracted: study name, year of data collection, type of study, location, gestational age, birth weight, age, ethnicity, BMI, dietary assessment method and validation, supplement use, and which foods, nutrients, or biochemical markers of nutrient status were reported.

For the current analyses, articles that reported data for pregnant women were selected, and data on food intake and nutrient intake were subsequently extracted. The extraction was restricted to the foods and nutrients mentioned in the guidelines and recommendations for pregnant women of the Dutch Health Council [6,9,10]. If available, data on nutrient intake, nutrient status, and the trimester at which the assessment took place were extracted. For data reported per subgroup (e.g., degree of adherence to the Mediterranean diet), a weighted median was calculated. Multiple articles refer to the same study population. To prevent duplicate data (i.e., data from the same study population), the article with the largest sample size was selected and included in the qualitative analysis. For fish and meat, exemptions were made. Instead of selecting the paper with the largest sample size, the most recent record of Stratakis et al. was selected, as it included one additional cohort and the interquartile range of fish consumption [20]. For meat intake, instead of the article with the largest sample size, an article that discriminated fresh and processed meats was selected [21]. The Dutch Health Council also included recommendations on food safety in their dietary recommendations for pregnant women [6]. Examples are the prevention of food infections and limiting the exposure to dioxins and lead. We did not include these aspects of the nutrition recommendations for pregnant women in the current review.

### 2.3. Evaluation

To gain insight into the adequacy of the intakes of pregnant women, the nutrient intake data were qualitatively compared to the Dutch dietary recommendations (see Table 1) [6] and reference values (see Table 2) [9,10] for pregnant women. It was assumed that the distribution of the nutrient requirements was normal, except for iron [22]. For the qualitative comparison with the reference values and recommendations, the intakes, as presented in the articles or reports, were adopted. In general, means with standard deviations or medians with a range were reported, with no additional information on the distribution. Therefore, it was not possible to estimate the proportion with inadequate intakes. In addition, food frequency questionnaires (FFQs) were used. The qualitative comparison therefore only provides a first indication of potential adequate intakes or too low intakes. Mean or median intakes above the EAR (estimated average requirement) or AI (adequate intake) were considered adequate. In cases where the mean or median intakes fell below the EAR, the intake was considered inadequate for a large proportion of the population. In cases where the mean or median intakes were above the EAR, the intake was additionally compared to the RDA (recommended daily allowance). No statement could be made when intakes were below the AI. To confirm the findings on the intake, information on nutrient status was subsequently evaluated. For vitamin C, the intake was only compared to the RDA, as no EAR was set by the Dutch Health Council [9]. Mean or median intakes were considered adequate when above the RDA, and no statement could be made when intakes were below the RDA. In addition to the qualitative comparison, results should be interpreted with care due to the quality of the data (e.g., assessment method, year of assessments; see Section 4.3. Discussion Quality of the data).

Information on intake, as reported in the articles, was compared with the dietary recommendations and reference values, except for protein. The estimated average requirement (EAR) for protein was set as grams per kilogram of body weight. However, the protein intake was reported in the articles as grams per day. Therefore, the EAR was recalculated based on a reference body weight. The EAR for protein for women aged 18–29 years was 0.66 g per kg of bodyweight [10]. Based on a reference weight of 64.6 kg, the EAR was 43 g protein per day [10]. Pregnant women require an additional 0.5 (first trimester), 7.2 (second trimester), and 23.0 (third trimester) g per day; 43.5, 50, and 66 g per day, respectively. RDA was calculated to be 54.6, 62.6, and 81.6 g per day. 

To interpret the nutrient status data, a qualitative comparison was made by comparing the mean/median status with cut-off values. The following cut-off values were used, and status was considered insufficient when mean/median status was below: for vitamin D (25-hydroxyvitamin D) status of 25 nmol/L [12]; serum or plasma folate levels of 6.8 nmol/L and red blood cell (RBC) folate levels of 906 nmol/L [23]; ferritin status of 15 µg/L (first trimester) [24]; urinary iodine-to-creatinine ratio (UI/Creat) concentration of 150 µg/g [25,26,27]. For vitamins B6 and B12, the status was considered adequate when the mean/median was within the reference range of 35–110 nmol/L or 130–700 pmol/L, respectively [28]. For copper and zinc status, no interpretation of insufficiency was made, as serum/plasma concentrations are considered of limited value for identifying status [29,30].

Figures were included if data were reported in more than two studies or included multiple subgroups (e.g., dietary and supplemental intake). In cases of nutrient status data, figures were included when intakes were below the EAR or AI. The reference software EndNote was used, and study characteristics, food and nutrient intake, and nutrient status were gathered in Microsoft Excel (version 2102). Figures were created using GraphPad PRISM (version 9.1.0).

## 3. Results

For an overview of the record selection, see the PRISMA flow diagram (Figure 1). The literature search resulted in 321 reports, including those identified through a previous search [14]. Of these, 218 reports contained data on pregnant women and were included in the current review. A total of 54 studies were included. For an overview of the study characteristics, see Appendix B, Table A3. The oldest study identified was the MEFAB (1989–1995), and the most recent study identified was APROPOS-II (2019–2021) [31]. The number of participants varied between 21 and up to 8742 women per study. The most often used dietary assessment method was the food frequency questionnaire (FFQ). Alcohol use and folic acid supplement use were most frequently reported and assessed via general questionnaires.

### 3.1. Dietary Pattern

Concerning the dietary pattern, data were found for fruit, vegetable, and fish intakes and alcohol use. Limited information (1–3 studies) was available for legumes, fats and oils, caffeine, sugary drinks, and salt intakes. No information on the consumption of unsalted nuts, wholemeal products, or red meat was found. The most recent data originated from 2021 (vegetables, fruit, alcohol, and caffeine), 2019 (plant-based diet), 2012 (fish), and 2005 (legumes, margarines and cooking fats, sugary drinks, and salt intakes) (see Appendix B, Table A3).

#### 3.1.1. Fruits and Vegetables

Most pregnant women did not reach the recommended intake for fruit and vegetables. The mean intake of vegetables ranged from 136 to 158 g per day (see Figure 2) [32,33]. About 23–35% of the women had an intake equal to or above the recommended 200 g per day [34,35]. Only half (51–56%) of the women consumed vegetables daily [36,37]. About 26–62% consumed at least two pieces of fruit daily [31,33,34,35,38]. The intake of fruit was 143 (mean) and 187 (median) grams per day [32,39].

#### 3.1.2. Fish

Stratakis et al. reported the fish intake of five Dutch cohort studies (i.e., ABCD, Generation R study, KOALA, LucKi, and PIAMA, see Figure 3) [20]. Pregnant women consumed fish 0.4 to 1.0 times per week [20,40], which is below the recommended 2 times per week. The median intake of fatty fish was 0.3 to 0.5 times per week [20,40,41]. The DHA intake was assessed in two studies: the mean intakes were 70 and 120 mg per day [42,43]. The mean EPA intake was 30 mg per day [43]. For those who do not consume fish, the recommendation is to use fish fatty acid supplements. One study reported on the use of fish oil supplements (including EPA and DHA): about 0.2% used these type of supplements [32]. It is, however, unknown whether these women did not consume fish.

#### 3.1.3. Alcohol

The frequency of alcohol consumption was often assessed in studies. The percentage of women using alcohol during pregnancy varied greatly. Most studies reported that up to a quarter (23%) of the women consumed alcohol. However, several studies reported higher percentages of 36–54% [44,45]. The duration of alcohol consumption during pregnancy was often unclear. Three studies indicated that women stopped or reduced their consumption as soon as the pregnancy was known. Beijers et al. reported that 33% of the women did not use alcohol at all during pregnancy, 61% stopped when pregnancy was known, and 5% continued consumption during pregnancy [46]. Poels et al. indicated that 26% of the women quit consumption before pregnancy was known, 62% quit after pregnancy was known, and 12% continued the consumption during pregnancy [47]. Gootjes et al. reported that 51% did not use alcohol during pregnancy, 13% stopped consumption when pregnancy was known, and 36% continued consumption during pregnancy [44]. A few studies reported on the quantity of alcohol consumed. Brinksma et al. reported that 81% were nonusers, 14% of the women consumed less than one glass per week, and 5% reported one glass or more [48]. Dirix et al. reported that 89% were nonusers and 11% consumed one glass or more per week [49]. Looman et al. reported daily consumption, with a significant difference between the trimesters [50]. Nonusers were 55%, 96%, and 92% at preconception, first trimester, and second trimester, respectively. The percentages of women consuming up to one glass per week were 39% at preconception, 4% at the first trimester, and 8% at the second trimester. The percentage of women consuming more than one glass of alcohol per day decreased from 6% at preconception to 0% at the first and second trimester.

#### 3.1.4. Legumes, Caffeine, Fats and Oils, and Sugary Drinks

Some studies reported on legume intake (up to a mean intake of 9 g per day) [32]. No data were available on the percentage of pregnant women consuming legumes weekly and conforming to the recommendation. Three cohort studies (ABCD, APROPOS II, and Generation R) reported the caffeine intake of pregnant women, of which two reported intakes below and above 200 mg [31,51,52]. As 58–67% of the women had an intake that was smaller than the maximum of 200 mg per day, a substantial part (33–42%) still exceeded the recommendation [51,52]. Only one study reported on the oil, margarine, and butter intake of pregnant women [21], indicating that women mainly consumed margarine (median of 15.7 g per day) and vegetable oil (median of 7.8 g per day). The median butter intake was 0 g per day (90% range of 0–17.8 g per day). Pregnant women consumed about two servings of sugary drinks per day (sugar-containing beverages including soda, fruit juice, and concentrate) [53]. The estimated salt intake was about 8 g per day [54], which exceeds the maximum limit of 6 g per day.

#### 3.1.5. Nuts, Wholemeal Products, and Meat 

It remains unclear whether pregnant women consume the recommended amount of unsalted nuts: only one study reported nut intake. The median intake of 18 g per day was above the recommended 15 g; however, the intake probably included salted nuts [21]. Three studies reported on the intake of cereal products; however, none contained details on the intake of refined or wholegrain products [21,32,39]. No information was available on the consumption of red meat. One study reported a lower median consumption of processed meat (25 g per day) compared to fresh meat (53 g per day) [21].

### 3.2. Nutrient Intake, Status, and Food Supplement Use

Nutrient intake data were available for protein, folate, vitamin B_12_, and calcium. Limited data were found for vitamin A, thiamine, riboflavin, niacin, vitamin B_6_, vitamin C, vitamin D, iron, and magnesium intakes. No intake data were found for vitamin K_1_, iodine, potassium, copper, and zinc. Data on nutrient status were found for vitamin B_6_, folate, vitamin B_12_, vitamin D, iron, iodine, copper, and zinc. Folic acid and vitamin D supplement use were frequently reported. The most recent data originated from 2019 (folic acid supplement use, and iodine), 2017 (protein, vitamin B_6_, folate, vitamin B_12_, vitamin D, and iron/ferritin), 2015 (vitamin D supplement use and calcium), 2014 (copper and zinc), 2010 (vitamin A, thiamin, riboflavin, and vitamin C), 2006 (nicotinamide), and 2005 (magnesium).

#### 3.2.1. Protein intake

Protein intake was assessed in four studies (see Figure 4) [50,55,56,57]. The mean/median intakes ranged from 75 to 88 g per day. These intakes were above the EARs and RDAs for the first and second trimesters. No information was available on the protein intake during the third trimester.

#### 3.2.2. Folic Acid

The folic acid intake was reported in three studies [43,50,57]. The median dietary folate intake ranged from 178 to 286 µg per day (see Figure 5) [43,50,57]. The mean dietary intakes were 286, 284, and 282 µg per day during preconception, the first trimester, and the second trimester, respectively [50]. The mean/median dietary folate intake was below the AI of 400 µg per day in dietary folate equivalents. The mean supplemental folic acid intake was 362 µg (preconception), 625 µg (first trimester), and 396 µg (second trimester) per day [50]. The supplemental intake during preconception was below the recommended 400 µg per day; the intake during the first trimester was above the recommendation. A large heterogeneity was seen in folic acid supplement use: 50–98% used folic acid supplements during pregnancy [31,34,38,45,47,56,57,58,59,60,61,62,63,64,65,66,67,68,69,70,71,72,73,74,75,76,77]. Seven studies reported the correct use of folic acid supplements: 46–71% of women used supplements at least 4 weeks prior to conception and up to 8 weeks after conception [45,59,64,68,72,75,77]. It is unclear whether they continued folic acid supplement use during the remainder of their pregnancy. The mean/median folate status was above the cut-off value of 6.8 nmol/L (see Figure 6). Looman et al. observed a significant increase in folate status from preconception (29.3 nmol/L) to the first trimester (41.1 nmol/L) and a significant decrease in the second trimester (29.7 nmol/L), which was associated with supplement use [50]. Two studies reported the folate status of supplement users and non-users separately, with a lower status among non-users. Folate status was above the cut-off value for both supplement users and non-users. The mean status was 20.8 nmol/L for users and 9.6 nmol/L for non-users [70], and the median status was 31.3 nmol/L for users versus 12 nmol/L for non-users in the first trimester [78]. The red blood cell folate status was 1408 nmol/L (median of 97% for supplement users) [79] and 1480 nmol/L (mean of 98% for supplement users) [57], which is above the cut-off value of 906 nmol/L. 

#### 3.2.3. Vitamin B_12_

Vitamin B_12_ intakes were reported in three studies [50,57,81] and were above the EAR of 2.4 µg per day and below and above the RDA of 3.3 µg per day. The mean/median dietary intakes were 3.1–5.0 µg per day (see Figure 7). The total mean intakes (including supplements) ranged from 6.6 to 8.8 µg per day. No significant differences were observed between the trimesters of pregnancy [50]. The mean/median vitamin B_12_ status ranged from 172 to 308 pmol/L [50,57,65,80,81,82]. The active vitamin B_12_ status was assessed in one study, with a median of 42 pmol/L [82]. Looman et al. reported a significant decrease in the mean vitamin B_12_ status during pregnancy: 308 pmol/L at preconception, 258 pmol/L in the first trimester, and 210 pmol/L in the second trimester [50]. The mean/median vitamin B_12_ statuses were within the reference range of 130–700 pmol/L.

#### 3.2.4. Calcium

Three studies reported the calcium intake of pregnant women (see Figure 8). The mean/median dietary calcium intakes were between 798 and 1145 mg per day [57,83,84]. The study populations had a mean age above 25 years, and the calcium intakes were above the age corresponding EAR of 750 mg per day and below and above the RDA of 950 mg per day. Based on these studies, not all women may reach the recommended adequate intake of 1000 mg per day at 20 weeks of gestation; 60% of the women had intakes below 1000 mg per day for up to 16 weeks of gestation [83]. Willemse et al. reported a total mean calcium intake of 950 mg per day [83]. Seventy percent of the women used a calcium-containing (prenatal) multivitamin supplement; two percent used a calcium-specific supplement. The median calcium intake from supplements was 395 mg per day.

#### 3.2.5. Vitamin A, Riboflavin, Niacin, and Vitamin B_6_

Vitamin A intake was assessed in one study [57]. The median intake was 877 mg retinol equivalent (RE), which is above the EAR of 580 mg RAE and the RDA of 750 mg RAE. The same study assessed the thiamine intake of pregnant women [57]. The median intake was 1.2 mg per day (about 0.137 mg/MJ) and exceeded the RDA. Riboflavin intake was assessed in two studies, and it was found to be below or equal to EAR, with median intakes of 1.4 (assessed 16 months after pregnancy) and 1.5 mg per day [43,57]. Data on nutrient status are needed to verify whether there is an insufficient intake. Niacin intake was assessed in one study [43]. The intake of 15 mg per day was at the RDA for the first trimester but not above the RDA for the second and third trimesters. The intake, however, did not include the niacin synthesis from tryptophan [85], and was assessed 16 months after pregnancy.

The total mean/median vitamin B_6_ intakes were above the EAR and RDA (see Figure 9) [50,57]. Looman et al. reported the intake of women in different trimesters. No significant differences were found in the total intake, dietary intake, and supplemental intake between the preconception, first, and second trimester periods. The mean vitamin B_6_ status was significantly lower in the second trimester (80.0 nmol/L) compared to preconception and first trimester levels (89.8 and 88.7 nmol/L, respectively); however, all were within the reference range of 35–110 nmol/L.

#### 3.2.6. Vitamin D

Vitamin D intake was reported in one study [50] (see Figure 10). The mean dietary vitamin D intake during preconception was 3.5 µg per day and 3.3 µg per day in the first and second trimesters. The mean total vitamin D intakes during preconception, the first trimester, and the second trimester were 7.7, 10.4, and 8.9 µg per day. The mean total intake during the first trimester was above the adequate intake of 10 µg per day. The increased intake in the first trimester was related to the supplemental intake [50]. Eight studies reported vitamin D supplement use [32,69,86,87,88,89,90]. Supplement use ranged from 3% (vitamin D-specific supplement) to 89% (including multivitamins), with about half of the women using a supplement with the recommended dosage of 10 µg [32,88,90]. One study reported on supplement use in more detail: 46% of pregnant women used a supplement containing vitamin D during pregnancy, of which 54% used vitamin D supplements throughout the entire duration of their pregnancy [86]. The supplement dose was in line with the recommendation: 97% used a multivitamin supplement containing 10 µg. The mean (46–89 nmol/L) or median (47–84 nmol/L) vitamin D status was above the reference value (see Figure 11, measured throughout the year) [50,88,91,92,93].

#### 3.2.7. Iron

The mean/median dietary iron intake was between 10.5 and 12.2 mg per day (see Figure 12) [50,57,94]. The iron intakes were above the EAR and below the RDA. As it cannot be assumed that the distribution of the iron requirement is normal, a comparison with the EAR will underestimate the risk of an inadequate intake [22]. Looman et al. reported an increase in the total iron intake during pregnancy due to an increased intake via supplements [50]. Three studies reported on iron supplement use, indicating that 18 and 36% of the women used iron-containing supplements during pregnancy [65,94,95]. The mean iron status was 17 and 22 µmol/L [65,96]. The mean/median ferritin status ranged from 12.8 to 52.2 µg/L [50,65,96]. Looman et al. reported a significant decrease in the ferritin status in the second trimester (12.8 µg/L) compared to preconception and the first trimester (31.7 and 31.4 µg/L, respectively) [50]. Ferritin status was above the reference value during preconception and the first trimester. As the WHO reference value is for the first trimester only, no evaluation of the status during the second trimester could be made. One study, on hematological parameters, identified that about 20% of their study population (in the third trimester of pregnancy) had a suspected latent iron deficiency [97].

#### 3.2.8. Vitamin C, Iron, and Magnesium

Only one study reported on vitamin C intake [57] and one on magnesium intake [98]. The median vitamin C intake was 102 mg per day and above the RDA. The mean magnesium intake was 339 mg per day, which is above the adequate intake of 300 mg per day.

#### 3.2.9. Vitamin K_1_, Iodine, Potassium, Copper, or Zinc

No data were found for the vitamin K_1_, iodine, potassium, copper, or zinc intakes of pregnant women. For iodine, copper, and zinc, several studies were found reporting the status data of these nutrients. Two cohort studies reported urinary iodine concentrations. Dineva et al. reported a median urinary iodine to creatinine ratio of 210 µg/g [26], which is above the cut-off value of 150 µg/L for insufficiency. Mayunga et al. reported, however, a median ratio of 141 µg/g, where 58% of the women had insufficient iodine concentrations [99]. A total of 40% of the women used iodine-containing supplements, of which 29% used a dose of 75 µg per day and 17% a dose of 150 µg per day [99]. Women who were not consuming iodine-containing supplements had a significantly lower urinary iodine to creatinine ratio (130 µg/g) than those consuming a 75 (148 µg/g) or 150 µg iodine supplement (171 µg/g) [99]. One study reported the copper and zinc status and intake of these micronutrients via supplements in pregnant women [87]. About 73% of women used a copper- and zinc-containing supplement during early pregnancy. No significant differences were found in the copper and zinc status for supplement users compared to non-users: the copper status was 26.29 µmol/L and 26.25 µmol/L, and the zinc status was 12.57 µmol/L and 12.55 µmol/L, respectively.

#### 3.2.10. Multivitamin Supplements

The Dutch Health Council recommends the use of folic acid and vitamin D supplements during pregnancy. The use of a supplement containing multiple vitamins and minerals is not recommended; however, it may be useful when a diet appears inadequate for several nutrients due to dietary restrictions (e.g., not consuming fish). Multivitamin supplement use was reported in several studies and ranged from 18% to 80% [56,58,69,71,74,76,84,87,91,100,101]. Prenatal-specific supplement intake increased during early pregnancy: 29% of the women started the use before pregnancy, and at 8 weeks of pregnancy, 61% used this type of supplement [102]. The general multivitamin use decreased from 8% before pregnancy to 5% at 8 weeks of gestation.

## 4. Discussion

This systematic review provides a comprehensive overview of the food and nutrient intakes of pregnant women in The Netherlands. In addition, it compares the intakes to the 2021 Health Council dietary recommendations and reference values for pregnant women.

Based on the current literature review, pregnant women living in The Netherlands seem to have adequate intakes of protein, vitamin A, thiamin, riboflavin, vitamin B_6_, folate, vitamin B_12_, vitamin C, iron, calcium, and magnesium. For folate and vitamin D, additional intake through supplements was needed to reach the recommended intake. The use of these supplements varied greatly, and the correct use (recommended dose and timing) may be improved. Calcium intake may not be sufficient after 20 weeks of gestation. About half of the women had a caffeine intake that was above the recommendation. The reasons for concern are the intakes of fruits, vegetables, and (fatty) fish, which were below the recommendations, and the intakes of alcohol, sugary drinks, and salt, which exceeded the recommendations. No data were available to evaluate the intake of unsalted nuts, weekly legume consumption, wholegrain cereal products, red and processed meat, niacin, vitamin K_1_, potassium, copper, zinc, and iodine.

### 4.1. Previous Research

Blumfield et al. published a meta-analysis on the dietary intake of micronutrients in developed countries [103]. In line with our findings, they concluded that there is an adequate intake of thiamin, riboflavin, niacin, vitamin B_12_, vitamin C, and calcium in most countries. In contrast, they found an inadequate intake of iron. This conclusion was, however, based on the EAR results, which were 2–3 times higher than the one used in the current review. As in our review, vitamin D and folate dietary intakes were found to be inadequate. Blumfield et al. found information on the vitamin A and zinc intakes of pregnant women in Italy, Finland, Sweden, the UK, Norway, and Spain. Intakes were above the EARs. As in our review, no data were found for iodine and vitamin K intakes.

Pregnant women did not adhere to the recommendations for fruit, vegetables, (fatty) fish, alcohol, sugary drinks, and salt. Similar findings were observed in the general Dutch population [104,105]. In the general population, changes were, however, seen over time, with a small increase in vegetable and fruit consumption, a small decrease in alcohol consumption, and a significant decrease in sugary drinks (2012–2016). In addition, the intake of legumes and unsalted nuts increased, and the intake of (red) meat and processed meat decreased. It is, however, unclear whether this trend over time is also seen among pregnant women. The estimated salt intake was about 8 g among pregnant women. This high intake is in line with the intake of the general Dutch population [106]. Recent data (2020–2021), based on 24 h urine excretion, indicated that women had a median intake of 8.5 g per day. Although the salt intake has decreased over the past 15 years, it still exceeds the recommendations. Reducing one’s salt intake will lower blood pressure; there is, however, insufficient evidence to support an effect for the prevention of pre-eclampsia [107,108]. 

The current review did not identify iodine intake data for pregnant women. One cohort reported the iodine status based on spot-urine analyses assessed in 2002–2006 [26]. The authors concluded that pregnant women were iodine-sufficient (median iodine-to-creatinine ratio of 210 µg/g, 25–75th percentiles: 140–303 µg/g). About 29% of women were below the urinary iodine-to-creatinine ratio cut-off value of 150 µg/g. A second study, however, assessed in 2018–2019, concluded that iodine status was insufficient (median iodine-to-creatinine ratio of 141 µg/g, range: 42–1938 µg/g), with 58% of women below 150 µg/g [99]. Based on a recent study among the general Dutch population, iodine intake was sufficient, but should not decrease any further [109]. The median intake among women decreased over the past 15 years (from 234 µg/d to 153 µ/d), which is similar to our findings for pregnant women. This decrease is, at least partly, the result of a change in the legislation in 2008, resulting in reductions in iodine fortification in bread [110]. Pregnant women have an increased iodine need for a properly functioning thyroid gland, which is important for the child’s growth and brain development [6]. The Iodine in Pregnant Women Study (Jodium in Zwangere vrouwen Onderzoek, JOZO) is currently ongoing to determine the iodine status among pregnant women in The Netherlands [111].

### 4.2. Supplement Use

Dietary vitamin D and folate intakes were inadequate; however, we also found that supplementation helped to fill the gap. This is in line with the current recommendation for pregnant women to use vitamin D and folate supplements. Adherence to these recommendations is, however, not optimal. Supplement use varied greatly among pregnant women. Folic acid supplement use ranged from about 50–100%, and vitamin D use ranged from 3–89%. Another finding was that folic acid supplements are often not used during the entire recommended period. These findings are in line with a recent publication on folic acid supplement use in The Netherlands [112]. Although the percentage of non-users was low (3%), women often did not follow the guidelines correctly (34%). Supplementation was started too late (92%), or women stopped too early (12%). The overall use improved over time (2014–2019); however, the correct use did not improve, indicating that knowledge about the importance of folic acid use improved but a better understanding of correct use is required. Women up to 25 years old with a low or middle education level, women not born in The Netherlands, and women not with a first pregnancy were identified as risk groups for incorrect folic acid supplement use. It is, however, unclear whether this resulted in an increased risk for neural tube defects. The possible reasons for not adhering to recommendations may be an overestimation of their own health, their perceived knowledge, or a possible underestimation of their health risks [31]. The knowledge and correct use of folic acid supplements were found to be higher among women with a higher education compared to those with a lower education [64]. For vitamin D, about half of the women did not use the recommended dosage. This may be due to the changed recommendation. In 2008, the Health Council thought it desirable to use vitamin D supplements; in 2012, supplementation was recommended as a precautionary, and in 2021, vitamin D supplementation was advised for all pregnant women [6,16,113].

The literature suggests that women often start prenatal multivitamin supplement use during early pregnancy [102]. Although the use of a multivitamin supplement is not generally recommended by the Health Council, women might choose such a supplement for practical reasons (i.e., containing both folic acid and vitamin D). In the case of multivitamin supplement use, the Health Council recommends the use of supplements specifically developed for use during pregnancy to ensure adequate dosages and prevent undesirable high dosages.

### 4.3. Quality of the Data

All dietary assessments were performed with FFQs. Although FFQs are often used in large-scale studies due to their relatively low costs and limited participant burden, they do not accurately assess the individual’s daily intake, but they are valid for ranking the subjects according to their intakes [114,115]. FFQs are often developed to cover the main dietary sources of a specific nutrient and include pre-specified foods. An example is the FFQ used by Willemse et al., which was specifically designed to assess calcium intake [102]. Based on their preselected foods, they were able to capture over 60% of the total dietary calcium intake, and adjustments were made in the analysis to reflect the total intake. This type of information on the performance of the FFQ is often lacking in publications, and FFQs are seldom validated in pregnant women. We identified only one study that used 24 h recalls, in which a semi-quantitative FFQ was validated against three non-consecutive 24 h recalls [57]. The study showed good validity for the FFQ assessment of folate and vitamin B_12_; however, validity was not assessed for other micronutrients and was unknown. The results of the 24 h recalls were not included in this review; however, based on the results of the 24 h recalls, the conclusions remained the same, except for riboflavin. Riboflavin intake was 1.7 mg/d above the EAR based on the 24 h recalls, whereas based on the FFQ intake, it was at the EAR (1.5 mg/d).

Dietary assessment methods are not always appropriate for assessing certain nutrients, such as salt intake [116]. Salt added during the preparation of the meal or at the table is difficult to assess accurately. Urine sodium excretion studies are therefore more appropriate for assessing the salt intake. The current review, however, did not identify such studies among pregnant women. It is desirable that future research include urine excretion assessments. In addition, research is needed on sensitive and specific biochemical nutrient status markers (e.g., copper and zinc status) in order to identify marginal deficiencies before the onset of severe deficiencies [29].

In addition to the methodology, the time span and national representativeness affect the usability of the identified studies for evaluating the nutrient intake of pregnant women in The Netherlands [14]. Dietary assessments were performed between 1989 and 2021 and were often limited to a few nutrients. For several nutrients, only a single study was identified with intake data, which makes the conclusions for these nutrients less strong. In addition, none of the studies were nationally representative of pregnant women living in The Netherlands. The time span may have influenced our findings, as recommendations differed at the time the dietary intake was assessed compared to the current 2021 recommendations. For example, the recommended fish consumption has increased from once per week (in 2015) to two times per week (2021) [6]. Another example is the use of vitamin D supplements, as described before. Recent data are therefore essential for evaluating the current recommendations.

### 4.4. Strengths and Limitations

It must be noted that the results should be interpreted with care. Most studies only reported the mean or median and did not provide insights into the intake or status distributions. As such, we had to base our evaluation on what was reported, resulting in a qualitative comparison of the mean/median with the EAR. For most nutrients, the proportion with an intake below the EAR is an estimate of the proportion with inadequate intakes. When the mean/median intake is below the EAR, a large part of the population is expected to have low intakes. When the mean/median intake is above the EAR, a proportion of the population might still be at risk of nutrient inadequacies. Additional information on the intake distribution is needed to make quantitative statements on nutrient inadequacy (i.e., calculation of the percentage falling below the EAR).

The literature search was designed and performed with great care. However, we might have missed certain publications due to publication bias or recent research that had not yet been published. In addition, we performed a general search on nutrient status and may have missed specific publications on biochemical nutrient markers of metabolic function (e.g., methylmalonic acid [117]). As the review includes over 200 articles, representing 50 (cohort) studies, we expect that our findings reflect the available nutritional data for pregnant women in The Netherlands.

### 4.5. Future Research

Our study showed that although there are several studies among pregnant women collecting data on nutrient or food intake or status, the reported data are often not sufficient for evaluating the dietary intakes and status of Dutch pregnant women. Therefore, the potential risks of nutrient inadequacies for mothers and their children are unknown, and no scientific-based advice or policies can be set. 

Based on the current findings, we are unable to propose effective intervention strategies and potential policies, as additional high-quality data and behavior research are needed. For several foods and nutrients, no intake data were found. It is important to assess these foods and nutrients and monitor their intake in order to evaluate the intakes and inform nutritional policies [103]. In addition, potential barriers and facilitators for adhering to the recommendations and guidelines should be studied in order to develop evidence-based interventions. Existing interventions and policies, such as for folic acid supplements, are currently being studied and evaluated. The ‘Power 4 a Healthy Pregnancy’ study is ongoing [118]. The diet quality will be assessed based on the 2021 Health Council guidelines, which will provide valuable information on, amongst others, legume, red/processed meat, and wholegrain product consumption. In addition, it will identify the potential barriers for pregnant women, midwives, and dietitians and study the effectiveness of counselling on diet quality.

The study by Looman et al. was the only study that measured nutrient intake and status at several timepoints during pregnancy and identified differences between these endpoints [50]. This shows that it is important to monitor intake and status during the entire pregnancy. In addition, this information will help improve nutrition recommendations during pregnancy, as the current recommendations are generally one value for the entire pregnancy due to a lack of information [9,24]. Physiological changes during pregnancy and inflammatory measures may affect nutritional status [24,50]. Better insight into these effects on the interpretation of nutritional status data is warranted [119]. In addition, monitoring and optimizing the nutrient intake of women of childbearing age is of importance, as the status of certain nutrients relies on the pre-pregnancy nutrient stores [103]. A proportion of Dutch women of childbearing age were found to have potential inadequate nutrient intakes [104], suggesting that their nutrient stores may be sub-optimal for pregnancy. The Dutch Health Council indicates that restricting the intake of animal-derived products may cause challenges in reaching certain recommendations and may increase the risk of inadequate intakes of fish (fatty acids), calcium, iron, iodine, and vitamin B_12_ [6]. In addition, the Health Council recommends vitamin B_12_ supplementation for those following a vegan diet. Data on vegetarian and vegan diets among pregnant women are, however, limited. Currently, almost 10% of Dutch women follow vegan or vegetarian diets [120]. Regarding the protein transition and the shift to a more plant-based diet, monitoring the diet, nutrient intake, and status is important.

A survey among midwives and obstetricians indicated that 60% and 24%, respectively, discussed maternal dietary preferences during their first prenatal consult [121]. However, they often considered their knowledge insufficient to provide advice on a strict plant-based diet. It is of interest to study how dietary preferences will evolve in the future and whether pregnant women are able to fill a potential gap through food choices, fortified foods, such as meat and dairy alternatives, or supplements, and whether they need additional advice. Midwives and obstetricians may receive additional training to increase their knowledge regarding plant-based diets during pregnancy.

There is a need for high-quality data. As mentioned before, representative and recent data are needed to evaluate the nutrient intake of pregnant women. The current review only provides a first indication of potential adequate intakes and too low intakes. A suggestion is to include multiple 24 h dietary recalls in future (cohort) studies or include pregnant women in food consumption surveys. This enables us to accurately estimate the habitual intake, determine the intake distribution, and quantify potential nutrient deficiencies. New technologies such as online 24 h dietary recalls and smart-phone food records may lower participant burden [122]. Subsequently, when using the same methodology, data from different (local) studies may be combined to form a (more) representative sample of pregnant women in The Netherlands. In addition, biochemical nutrient status markers may be assessed in a (sub) population to complement (e.g., urinary sodium and iodine) or confirm findings from the dietary assessment.

## 5. Conclusions

The current review identified several studies among pregnant women collecting data on nutrient intake, food intake, or status of Dutch pregnant women. However, the reported data are often not sufficient for evaluating dietary intake and status due to the methodology, time span, and national representativeness. For several foods and nutrients, no or limited intake data were found. Based on the available literature, pregnant women living in The Netherlands seem to have adequate intakes of protein, vitamin A, thiamin, riboflavin, vitamin B_6_, folate, vitamin B_12_, vitamin C, calcium, and magnesium. For folate and vitamin D, additional intake through supplements was needed to reach the recommended intake. The use of these supplements varies greatly, and the correct use (recommended dose and timing) may be improved. Calcium intake may not be sufficient after 20 weeks of gestation. About half of the women had a caffeine intake that was above the recommendation. The reasons for concern are intakes of fruits, vegetables, and (fatty) fish that are below the recommendations and intakes of alcohol, sugary drinks, and salt that exceed the recommendations. There is a need for high-quality, representative, and recent nutritional research in order to make accurate assessments and evaluate nutrient intakes, supporting scientific-based advice and national nutritional policies.

## Figures and Tables

**Figure 1 nutrients-15-03071-f001:**
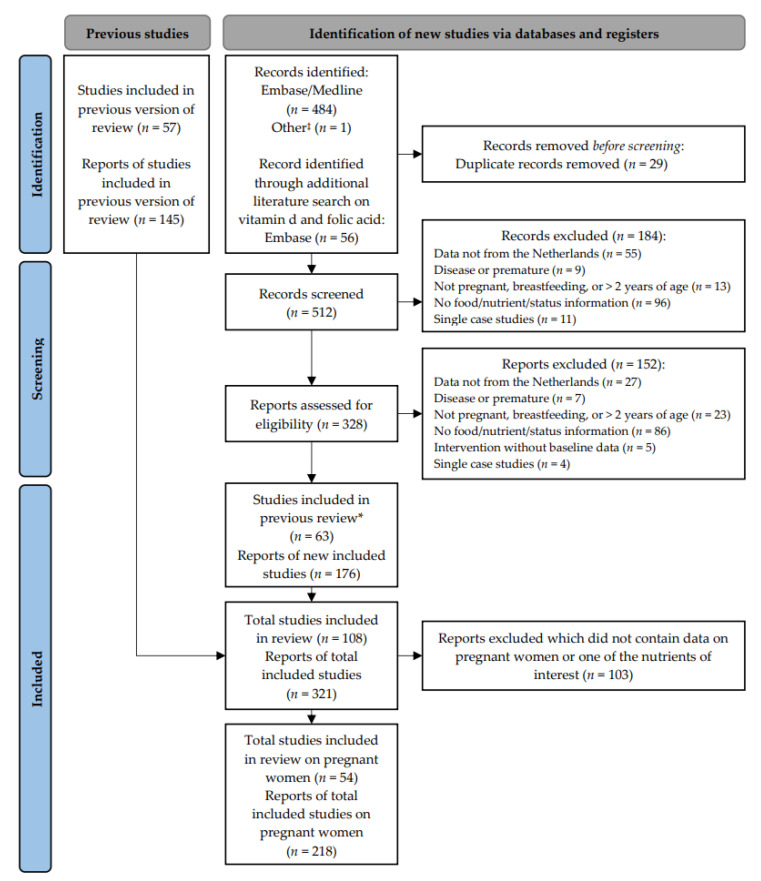
PRISMA 2020 flow diagram for updated systematic reviews [15]. ‡ Reports from Dutch institutes and non-peer-reviewed Dutch articles. * Studies that were already included in the previous review [14].

**Figure 2 nutrients-15-03071-f002:**
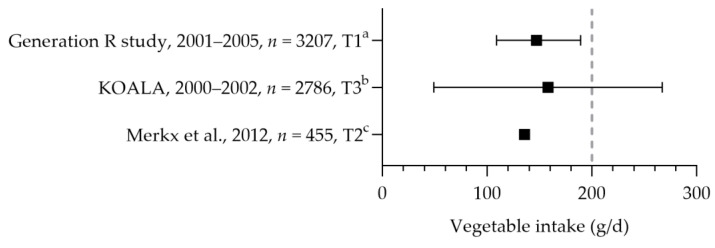
Vegetable consumption by pregnant women. Squares indicate vegetable intake. The dotted vertical line represents recommended intake (200 g/d). a = median with interquartile range (IQR); b = mean with standard deviation; c = mean. Generation R study [39]. KOALA [32]. Merkx et al. [33].

**Figure 3 nutrients-15-03071-f003:**
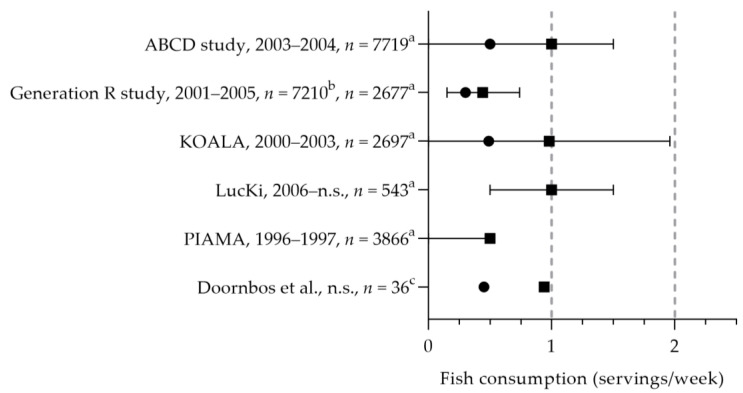
(Fatty) fish consumption by pregnant women. Squares indicate fish intake; dots indicate fatty fish intake. The dotted vertical line represents recommended intakes for fish (2 servings/week) and fatty fish (1 serving/week). a = median with interquartile range (IQR); b = median; c = mean. n.s. = not stated. ABCD study, Generation R study, KOALA, LucKi, PIAMA [20]. Doornbos et al. [40].

**Figure 4 nutrients-15-03071-f004:**
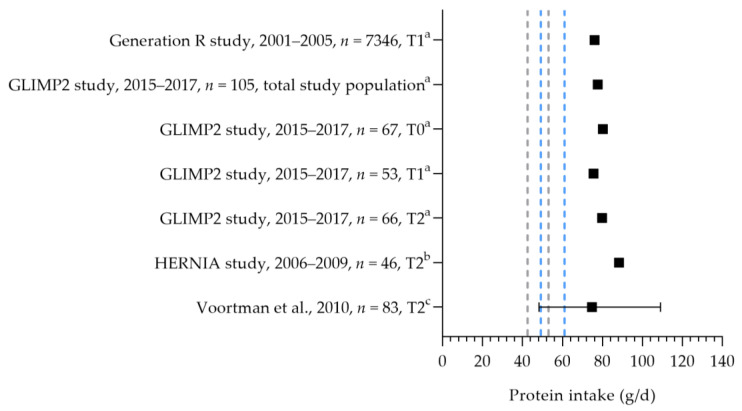
Protein intake of pregnant women. Squares indicate protein intake. The dotted lines represent the estimated average requirement (EAR: 43.5 g/d, 50 g/d) and recommended dietary allowance (RDA: 54.6 g/d, 62.6 g/d) for the first trimester (in gray) and second trimester (in blue). T0 = preconception; T1 = first trimester; T2 = second trimester. a = mean; b = median; c = median with 90% range. Generation R study [55]. GLIMP2 study [50]. Hernia study [56]. Voortman et al. [57].

**Figure 5 nutrients-15-03071-f005:**
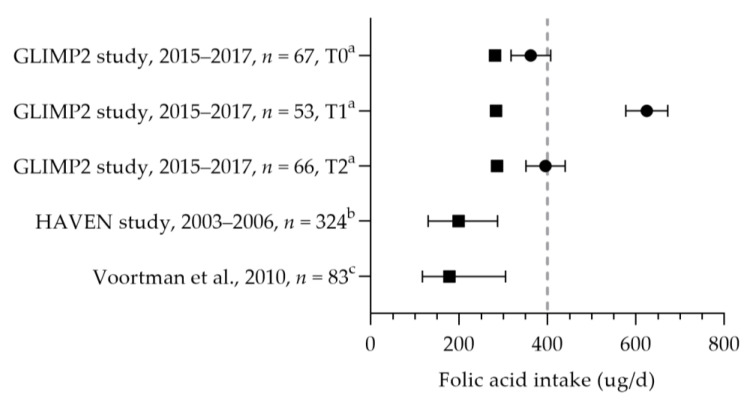
Folic acid intake of pregnant women. Squares indicate the dietary folate intake; dots indicate the intake via supplements. The dotted vertical lines represent the adequate intake (AI: 400 µg DFE/d) and the recommended intake from supplements (400 µg DFE/d one month before till the 10th week of pregnancy). T0 = preconception; T1 = first trimester; T2 = second trimester. a = mean with standard error of the mean (SEM); b = median with 5th and 95th percentiles; c = median with 90% range. GLIMP2 study [50]. HAVEN study [43]. Voortman et al. [57].

**Figure 6 nutrients-15-03071-f006:**
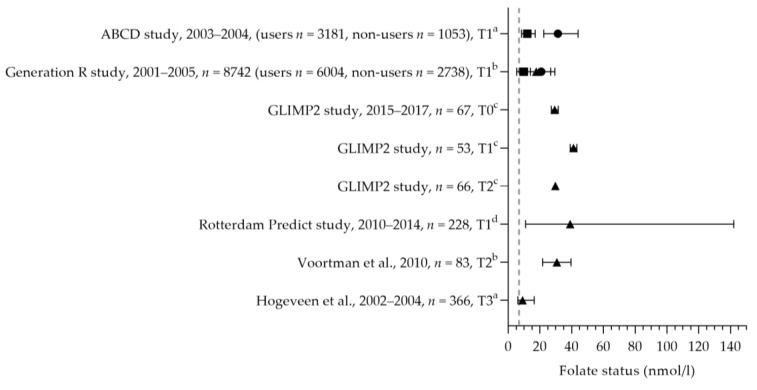
Folate status of pregnant women. Squares indicate the status of non-supplement users; dots indicate the status of supplement users; triangles indicate the status of the total study population. The dotted vertical line represents the reference value. T0 = preconception; T1 = first trimester; T2 = second trimester. a = median with interquartile range (IQR); b = mean with standard deviation (SD); c = mean with standard error of the means (SEM); d = median with minimum and maximum value. ABCD study [78]. Generation R study [70]. GLIMP2 study [50]. Rotterdam Predict study [80]. Voortman et al. [57]. Hogeveen et al. [61].

**Figure 7 nutrients-15-03071-f007:**
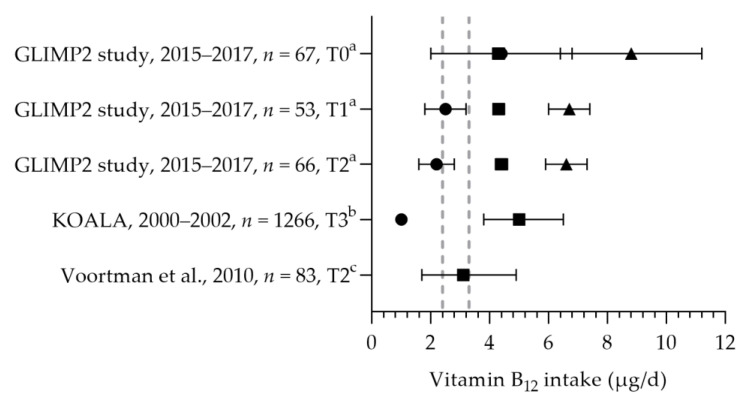
Vitamin B_12_ intake of pregnant women. Squares indicate the dietary folate intake; dots indicate the intake via supplements; triangles indicate the total intake. The dotted vertical lines represents the estimated average requirement (EAR: 2.4 μg/d) and the recommended dietary allowance (RDA: 3.3 μg/d). T0 = preconception; T1 = first trimester; T2 = second trimester. a = mean with standard error of the mean (SEM); b = median with interquartile range (IQR); c = median with 90% range. GLIMP2 study [50]. KOALA [81]. Voortman et al. [57].

**Figure 8 nutrients-15-03071-f008:**
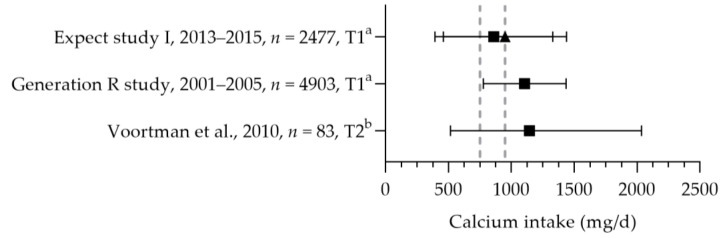
Calcium intake of pregnant women. Squares indicate the dietary folate intake; triangles indicate the total intake. The dotted vertical lines represent the estimated average requirement (EAR: 750 mg/d) and recommended daily allowance (RDA: 950 mg/d) for women aged 25 year and older. a = mean with standard deviation (SD); b = median with 90% range. Expect study I [83]. Generation R study [84]. Voortman et al. [57].

**Figure 9 nutrients-15-03071-f009:**
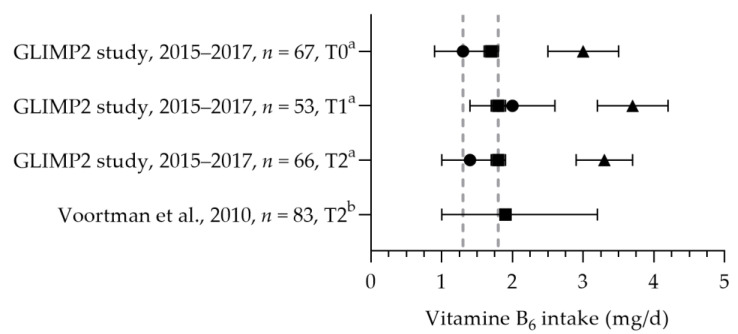
Vitamin B_6_ intake of pregnant women. Squares indicate the dietary folate intake; dots indicate the intake via supplements; triangles indicate the total intake. The dotted vertical lines represent the estimated average requirement (EAR: 1.3 mg/d) and recommended daily allowance (RDA: 1.8 mg/d). T0 = preconception; T1 = first trimester; T2 = second trimester. a = mean with standard error of the mean (SEM); b = median with 90% range. GLIMP2 study [50]. Voortman et al. [57].

**Figure 10 nutrients-15-03071-f010:**
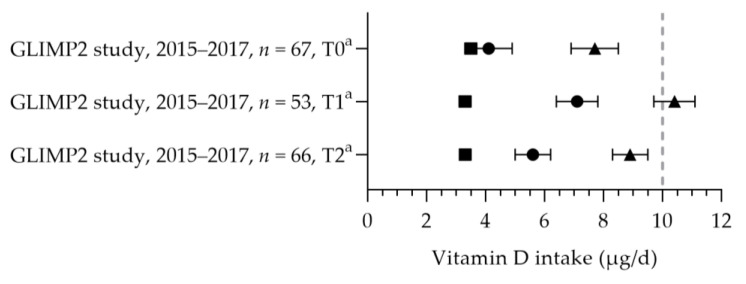
Vitamin D intake of pregnant women. Squares indicate the dietary folate intake; dots indicate the intake via supplements; triangles indicate the total intake. The dotted vertical line represents the adequate intake (AI: 10 μg/d). T0 = preconception; T1 = first trimester; T2 = second trimester. a = mean with standard error of the mean (SEM). GLIMP2 study [50].

**Figure 11 nutrients-15-03071-f011:**
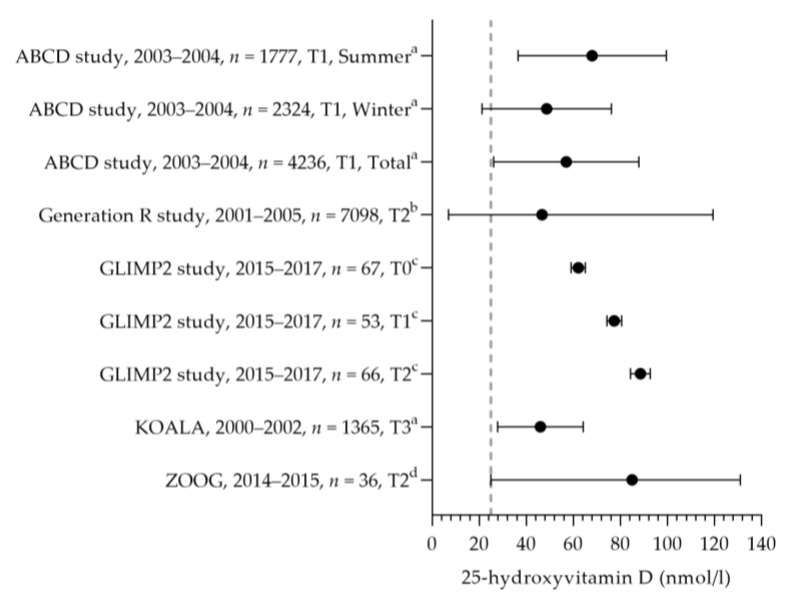
Vitamin D status of pregnant women. Dots indicate the vitamin D status. The dotted vertical line represents the reference value. T0 = preconception; T1 = first trimester; T2 = second trimester. a = mean with standard deviation (SD); b = median with 95% range; c = mean with standard error of the means (SEM); d = median with minimum and maximum value. ABCD study [93]. Generation R study [92]. GLIMP2 study [50]. KOALA [91]. ZOOG [88].

**Figure 12 nutrients-15-03071-f012:**
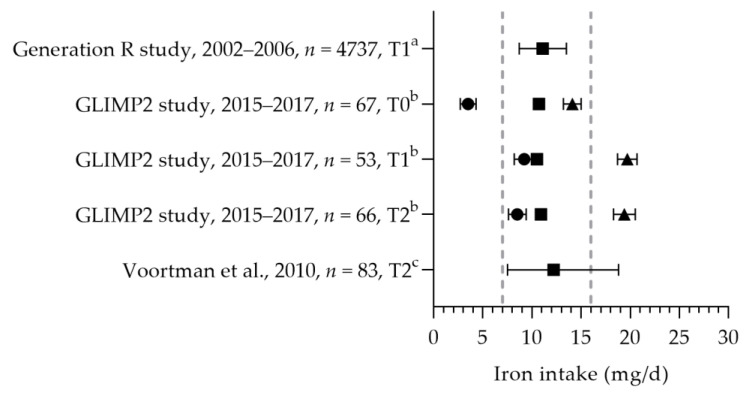
Iron intake of pregnant women. Squares indicate the dietary folate intake; dots indicate the intake via supplements; and triangles indicate the total intake. The dotted vertical lines represent the estimated average requirement (EAR: 7 mg/d) and recommended daily allowance (RDA: 16 mg/d). T0 = preconception; T1 = first trimester; T2 = second trimester. a = median with 25th and 75th percentiles; b = mean with standard deviation (SD); c = median with 90% range. Generation R study [94]. GLIMP2 study [50]. Voortman et al. [57].

**Table 1 nutrients-15-03071-t001:** The Dutch Health Council dietary recommendations for pregnant women [6].

Dietary Recommendations
Healthy and varied food Eat plenty of vegetables (at least 200 g), fruits (at least 200 g), and unsalted nuts (at least 15 g) every day. Eat legumes every week. Substitute refined cereal products with wholemeal products as much as possible. Substitute butter, hard margarine, and cooking fats with soft margarine, liquid cooking fats, and vegetable oils. Limit the consumption of red meat and especially processed meat. Drink as few sugary drinks as possible. Limit the intake of table salt to a maximum of 6 g per day. Weight gain and calorie requirements The committee makes no recommendation on the optimal weight gain during pregnancy. Fish and fish fatty acids Eat fish twice a week, including one serving of fatty fish and one serving of lean fish, picking fish species that do not contain excessively high levels of harmful substances. For women who cannot or do not want to eat this amount of fish, take fish fatty acid supplements containing 250 to 450 mg of DHA per day. Calcium-rich products Eat enough calcium-rich products to reach at least the dietary reference value of calcium. If the intake is consistently too low, take a supplement containing 1000 mg of calcium a day, starting from the 20th week of pregnancy. Iron-rich products Eat enough iron-rich products. Iodine-rich products Eat enough iodine-rich products to meet the dietary reference value of 200 micrograms of iodine per day. If you struggle to consistently consume enough iodine, take a supplement with up to 200 micrograms of iodine. Beverages Avoid alcohol. Do not take more than 200 mg of caffeine per day. Nutrient supplements Take a supplement containing 400 micrograms of folic acid a day, starting from at least four weeks prior to conception up to the 10th week of pregnancy (i.e., 8 weeks after conception). Take a supplement containing 10 micrograms of vitamin D per day. If the diet appears inadequate on several fronts, multi-vitamin and multi-mineral supplements may be practical alternatives. It is important to choose a supplement with dosages that are suitable for pregnancy.

**Table 2 nutrients-15-03071-t002:** Dietary reference values for pregnant women per day [9,10].

Nutrient	EAR	RDA	AI
Protein (g/kg)	0.66 + First trimester: 0.5 g Second: 7.2 g Third: 32 g	0.83 + First trimester: 1 g Second: 9 g Third: 28 g	
Vitamin A (μg RAE) ^1^	580	750	
Thiamin (B_1_ and mg/MJ)	0.072	0.1 (1.0 mg/d) First trimester: 0.9 mg/d Second: 1.0 mg/d Third: 1.1 mg/d	
Riboflavin (B_2_ and mg)	1.5	1.9	
Niacin (B_3_, mg, and NE/MJ)	1.3	1.6 (16 mg NE/d) First trimester: 15 mg NE/d Second: 16 mg NE/d Third: 17 mg NE/d	
Vitamin B_6_ (mg)	1.3	1.8	
Folate (μg DFE) ^2^			400
Vitamin B_12_ (μg)	2.4	3.3	
Vitamin C (mg)		85	
Vitamin D (μg)			10
Vitamin K_1_ (μg)			70
Calcium (mg)	<20 weeks gestation: 18–24 year: 860 ≥25 year: 750	<20 weeks gestation: 18–24 year: 1.000 ≥25 year: 950	≥20 weeks gestation: 1.000
Iron (mg)	7	16	
Iodine (μg)			200
Potassium (g)			3.5
Copper (mg)	0.8	1.0	
Magnesium (mg)			300
Zinc (mg)	7.0	9.1	

EAR = estimated average requirement; RDA = recommended daily allowance; RAE = retinol activity equivalents: 1 μg RAE = 1 μg retinol = 12 μg β-carotene = 24 μg other carotenoids; RE retinal equivalents: 1 μg RAE 1 = 1 μg retinol = 6 μg β-carotene = 12 μg other carotenoids; MJ = megajoules; NE = niacin equivalents: 1 mg NE = 1 mg niacin = 60 mg tryptophan; DFE = dietary folate equivalents: 1 μg DFE = 0.6 μg folic acid from fortified foods or supplement combined with food = 0.5 μg folic acid from supplements taken on an empty stomach. ^1^ For the dietary reference values, RAE and RE are interchangeable [11,12]. ^2^ With an additional 400 μg from supplements four weeks prior to conception up to the 10th week of pregnancy.

## Data Availability

No new data were created or analyzed in this study. Data sharing is not applicable to this article.

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
