# Peer review of "An Evaluation of Food and Nutrient Intake among Pregnant Women in The Netherlands: A Systematic Review"

_nutrients, 2023, doi:10.3390/nu15133071_

Round 1

Reviewer 1 Report

Dear All,

 Having analysed the paper entitled “Evaluation of food and nutrient intake among pregnant women in the Netherlands: a systematic review”, I reached the following conclusions:

1)     The content of the work is consistent with the aim of the work.

2)     The work addresses an important issue related to nutrition of pregnant females.

3)     The authors might want to formulate one review aim instead of two.

4)     The review methodology does not raise any objections.

5)     The results were presented in a clear and understandable way.

6)     The Discussion section was organized properly.

7)     It might be advisable for the authors to narrow down the directions of future research.       The authors might also want to move some excerpts from the current version of the review to the Discussion section.

8)     Conclusions correspond to the collected research material.

9)     The research literature was correctly selected.

10)  The presented tables are legible and do not raise any doubts.

Author Response

Dear reviewer,

Thank you for your interest and for taking the time to review our manuscript. Your comments and suggestions are very valuable and have helped us to strengthen our review.

Please find our response on your specific comments below in blue font. The alterations in the manuscript are indicated with track-changes.

Kind regards

Dear All,

 Having analysed the paper entitled “Evaluation of food and nutrient intake among pregnant women in the Netherlands: a systematic review”, I reached the following conclusions:

1)     The content of the work is consistent with the aim of the work.

2)     The work addresses an important issue related to nutrition of pregnant females.

3)     The authors might want to formulate one review aim instead of two.

We have formulated one aim (lines 85-88).

4)     The review methodology does not raise any objections.

5)     The results were presented in a clear and understandable way.

6)     The Discussion section was organized properly.

7)     It might be advisable for the authors to narrow down the directions of future research.       The authors might also want to move some excerpts from the current version of the review to the Discussion section.

Thank you for your suggestion for our future research. As we included both the dietary pattern and the vitamins and minerals, the manuscript is extensive. We have adapted the discussion section to make it more concise. We however think that topics of the results section indeed belong to this section and cannot be moved to the discussion. But perhaps we misunderstood your commend.

8)     Conclusions correspond to the collected research material.

9)     The research literature was correctly selected.

10)  The presented tables are legible and do not raise any doubts.

Reviewer 2 Report

The manuscript entitled “Evaluation of food and nutrient intake among pregnant women in the Netherlands: a systematic review” provides an updated overview of the available food and nutrient intake data of pregnant women in the Netherlands. The study presents a comprehensive summary about the nutrient intake of pregnant women. The meta-analysis was also performed whenever possible. However, the used method is not common for me.

1. In lings 11-12, “Articles were selected when published since 2008 and containing data on food consumption, nutrient intake or status of healthy pregnant women.” should be placed after “Embase, MEDLINE and national institute databases were used.”

2. In lines 61-84, the search strategies is puzzling. Why the search strategies have to link with the previous studies, since their PICO items are different. One targets for pregnant women while the other targets the first 1000 days of life. Is there other analysis supporting this method?

3. In figure 1, it is surprising to find only 1 study was found in MEDLINE.

4. In figure 1, what does Studies included in review mean?

5. The language should be improved.

Moderate editing of English language required

Author Response

Dear reviewer,

Thank you for your interest and for taking the time to review our manuscript. Your comments and suggestions are very valuable and have helped us to strengthen our review.

Please find our response on your specific comments below in blue font. The alterations in the manuscript are indicated with track-changes.

Kind regards

The manuscript entitled “Evaluation of food and nutrient intake among pregnant women in the Netherlands: a systematic review” provides an updated overview of the available food and nutrient intake data of pregnant women in the Netherlands. The study presents a comprehensive summary about the nutrient intake of pregnant women. The meta-analysis was also performed whenever possible. However, the used method is not common for me.

  1. In lings 11-12, “Articles were selected when published since 2008 and containing data on food consumption, nutrient intake or status of healthy pregnant women.” should be placed after “Embase, MEDLINE and national institute databases were used.”

We have moved these lines so that the databases are mentioned first (lines 12-14).

  1. In lines 61-84, the search strategies is puzzling. Why the search strategies have to link with the previous studies, since their PICO items are different. One targets for pregnant women while the other targets the first 1000 days of life. Is there other analysis supporting this method?

The current study is a continuation of our study from 2019 (ter Borg, S., et al. (2019). "Food Consumption, Nutrient Intake and Status during the First 1000 days of Life in the Netherlands: a Systematic Review." Nutrients 11(4). As indicated in lines 84-85 it concerns a living systematic review which is updated monthly. It is a literature search targeting the 1st 1000 days. For the current manuscript we focused on one target group within these 1ste 1000 days: pregnant women. We chose to describe the PICO of the main literature search (i.e. 1ste 1000 days) as this is in line with the search strategy we used. We subsequently selected papers with information on pregnant women (see section 2.2. Screening and extraction). This is also visualized in figure 1. To emphasize this we have added additional explanation in lines 103-104.

  1. In figure 1, it is surprising to find only 1 study was found in MEDLINE.

The Embase database we have used includes records from the MEDLINE database (lines 95-96). We identified 484 records through this database. The 1 record which was indicated as ‘Other’ concerns an article from a non-peer reviewed Dutch journal (i.e. VoedingNu). We have adapted figure 1 and the footer.

  1. In figure 1, what does “Studies included in review” mean?

It concerns the studies which were already included in our review from 2019. We have adapted the figure 1 and it’s footnote to explain this.

  1. The language should be improved.

Due to the indicated ‘moderate editing required’ and the limited time before re-submission, no extensive English revision was made by a native-speaker.

Reviewer 3 Report

There are two aims to the current systematic review: i) to update the overview of available 5data on nutritional intake and status among pregnant women in the Netherlands and ii) to determine whether their intake is in line with the Dutch dietary recommendations and reference values.

We are faced with a strong and robust systematic review paper, following the PRISMA flow diagram, mapping out the number of records identified, included, and excluded, and the reasons for exclusions. Nevertheless, we present the following observations:

Introduction

- To reinforce the relevance of the research, we ask authors to give an overview of the development of a range of indicators concerning the number of births and fertility rate in Netherlands, during the past 15 years (2008-2023).

- Authors should better contextualize the main drivers and the framework of the Dutch dietary recommendations.

Material and Methods

- We suggested that authors include the Dutch dietary recommendations (Table 1) and the specific dietary recommendations and dietary reference values for pregnant women (Table 2) in the Introduction section.

- For the Exclusion criteria for the parallel search, can authors clarify why they excluded women with a mean age younger than 20 years?

Discussion

- As this manuscript revealed whether pregnant women living in the Netherlands comply with the Dutch dietary recommendations and reference values specifically for pregnant women, authors should use these insights for proposing effective intervention strategies and policies to prevent potential health risks.

- Can authors explain why no data was available to evaluate the intake of unsalted nuts, weekly legume consumption, wholegrain cereal products, red and processed meat, niacin, vitamin K1, potassium, copper, zinc, and iodine?

- Do authors think that the pandemic COVID-19 had changed the dietary pattern of the population, and pregnant women in particular, as the period in study include the pandemic crises?

Author Response

Dear reviewer,

Thank you for your interest and for taking the time to review our manuscript. Your comments and suggestions are very valuable and have helped us to strengthen our review.

Please find our response on your specific comments below in blue font. The alterations in the manuscript are indicated with track-changes.

Kind regards

Comments and Suggestions for Authors

There are two aims to the current systematic review: i) to update the overview of available 5data on nutritional intake and status among pregnant women in the Netherlands and ii) to determine whether their intake is in line with the Dutch dietary recommendations and reference values.

We are faced with a strong and robust systematic review paper, following the PRISMA flow diagram, mapping out the number of records identified, included, and excluded, and the reasons for exclusions. Nevertheless, we present the following observations:

Introduction

- To reinforce the relevance of the research, we ask authors to give an overview of the development of a range of indicators concerning the number of births and fertility rate in Netherlands, during the past 15 years (2008-2023).

We have added information concerning the births and fertility in lines 75-77. 2022/2023 data will be published later this year and were therefore not included.

- Authors should better contextualize the main drivers and the framework of the Dutch dietary recommendations.

Additional information was added in lines 47-52 of the introduction, providing more context to the new guidelines for pregnant women.

Material and Methods

- We suggested that authors include the Dutch dietary recommendations (Table 1) and the specific dietary recommendations and dietary reference values for pregnant women (Table 2) in the Introduction section.

Table 1 and Table 2 are now included in the introduction.

- For the Exclusion criteria for the parallel search, can authors clarify why they excluded women with a mean age younger than 20 years?

This was based on the age range at which women most frequently have children (20-40 years) in the Netherlands (CBS report: Vruchtbaarheid aan het begin van de 21e eeuw, 2017). The exclusion criteria concerns the mean age of the study population, (slightly) younger women are therefore also included. We have checked the selection process and we didn’t exclude articles based on this criteria. It must however be noted that it’s of interest to study the young pregnant women population, especially those with unplanned pregnancies. In addition, teenage pregnancies may have specific nutritional needs and need different recommendations as stated by the Dutch Health Council. Specific recommendations for teenage pregnancies are not set by the Dutch Health Council. Future research focusing on the needs of this specific group of pregnant women is of interesting.

Discussion

- As this manuscript revealed whether pregnant women living in the Netherlands comply with the Dutch dietary recommendations and reference values specifically for pregnant women, authors should use these insights for proposing effective intervention strategies and policies to prevent potential health risks.

Effective interventions and policies are indeed essential. We however are not able to make such recommendations based on the current review. As indicated in the discussion, we need additional high quality data and behavior research to identify potential barriers and facilitators for adhering to the recommendations and guidelines in order to develop effective intervention strategies. Please note that within our institute we perform research concerning this topic. We have added explanation in section 4.5 Future research, lines 662-669.

- Can authors explain why no data was available to evaluate the intake of unsalted nuts, weekly legume consumption, wholegrain cereal products, red and processed meat, niacin, vitamin K1, potassium, copper, zinc, and iodine?

No literature was available on these topics. Much of the data identified was obtained before the publication of certain recommendations for specific food groups (lines 630-631). This might be the reason why details on for instance the type of nuts or meat were not of interest and not studied.

Certain nutrients are difficult to assess via food frequency questionnaires, or are burdensome for the participants (such as urine collection for sodium, potassium and iodine analyses). The need for sensitive and specific biochemical markers is indicated in lines 621-623. Another reason could be that these nutrients were not of interest for researchers. We therefore conclude that there is a need for good quality data, including the nutrients and food groups currently missing. In addition, with recommendations changing, it is required to monitor frequently.

- Do authors think that the pandemic COVID-19 had changed the dietary pattern of the population, and pregnant women in particular, as the period in study include the pandemic crises?

Based on our review we cannot conclude whether there was a change in dietary pattern of pregnant women during the COVID-19 pandemic. One of the included studies (reference 31: Maas et al. 2022 BMC Pregnancy Childbirth) was conducted partly during the pandemic, the authors however did not study or discuss this topic. All other studies we included had data collection prior to the COVID-19 pandemic. We can imagine that the COVID-19 pandemic must have affected dietary patterns, and lifestyle in general, for a proportion of the population. It is however difficult to say whether this affected the dietary pattern in a positive or negative way. In addition, the effect may differ between subjects as well as between specific nutrients and food groups. As only one study had data collection in this period, and as the results of this study were comparable with other studies, we did not include this in our review. For future research it would be interesting to study how the COVID-19 pandemic may have changed dietary patterns and to study in addition if these potential changes remained in the years after the pandemic.